# NAOMI: Non-Autoregressive Multiresolution Sequence Imputation

**Yukai Liu**
Caltech
yukai@caltech.edu

**Rose Yu**
Northeastern University
roseyu@northeastern.edu

**Stephan Zheng**[*]
Caltech, Salesforce
stephan.zheng@salesforce.com

**Eric Zhan**
Caltech
ezhan@caltech.edu

**Yisong Yue**
Caltech
yyue@caltech.edu

## Abstract

Missing value imputation is a fundamental problem in spatiotemporal modeling, from motion tracking to the dynamics of physical systems. Deep autoregressive models suffer from error propagation which becomes catastrophic for imputing long-range sequences. In this paper, we take a *non-autoregressive* approach and propose a novel deep generative model: **N**on-**A**ut**O**regressive **M**ultiresolution **I**mputation (`NAOMI`) to impute long-range sequences given arbitrary missing patterns. `NAOMI` exploits the multiresolution structure of spatiotemporal data and decodes recursively from coarse to fine-grained resolutions using a divide-and-conquer strategy. We further enhance our model with adversarial training. When evaluated extensively on benchmark datasets from systems of both deterministic and stochastic dynamics. In our experiments, `NAOMI` demonstrates significant improvement in imputation accuracy (reducing average error by 60% compared to autoregressive counterparts) and generalization for long-range sequences.

## 1 Introduction

The problem of missing values often arises in real-life sequential data. For example, in motion tracking, trajectories often contain missing data due to object occlusion, trajectories crossing, and the instability of camera motion [1]. Missing values can introduce observational bias into training data, making the learning unstable. Hence, imputing missing values is of critical importance to the downstream sequence learning tasks. Sequence imputation has been studied for decades in statistics literature [2, 3, 4, 5]. Most statistical techniques are reliant on strong assumptions on missing patterns such as missing at random, and do not generalize well to unseen data. Moreover, existing methods do not work well when the proportion of missing data is high and the sequence is long.

Recent studies [6, 7, 8, 9] have proposed to use deep generative models for learning flexible missing patterns from sequence data. However, all existing deep generative imputation methods are *autoregressive*: they model the value at cur-

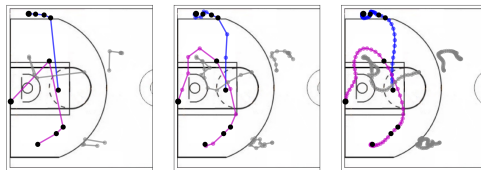

Figure 1: Imputation process of `NAOMI` in a basketball play given two players (purple and blue) and 5 known observations (black dots). Missing values are imputed recursively from coarse resolution to fine-grained resolution (left to right).

---

[*]This work was done while the author was at Caltech.

rent timestamp using the values from previous time-steps and impute missing data in a sequential manner. Hence, autoregressive models are highly susceptible to compounding error, which can become catastrophic for long-range sequence modeling. We observe in our experiments that existing autoregressive approaches struggle on sequence imputation tasks with long-range dynamics.

In this paper, we introduce a novel *non-autoregressive* approach for long-range sequence imputation. Instead of conditioning only on the previous values, we model the conditional distribution on both the history and the (predicted) future. We exploit the multiresolution nature of spatiotemporal sequence, and decompose the complex dependency into simpler ones at multiple resolutions. Our model, Non-autoregressive Multiresolution Imputation (`NAOMI`), employs a divide and conquer strategy to fill in the missing values recursively. Our method is general and can work with various learning objectives. We release an implementation of our model as an open source project.[2]

In summary, our contributions are as follows:

- We propose a novel non-autoregressive decoding procedure for deep generative models that can impute missing values for spatiotemporal sequences with long-range dependencies.
- We introduce adversarial training using the generative adversarial imitation learning objective with a fully differentiable generator to reduce variance.
- We conduct exhaustive experiments on benchmark sequence datasets including traffic time series, billiards and basketball trajectories. Our method demonstrates $60\%$ improvement in accuracy and generates realistic sequences given arbitrary missing patterns.

## 2   Related Work

**Missing Value Imputation**   Existing missing value imputation approaches roughly fall into two categories: statistical methods and deep generative models. Statistical methods often impose strong assumptions on the missing patterns. For example, mean/median averaging [4], linear regression [2], MICE [10], and k-nearest neighbours [11] can only handle data missing at random. Latent variables models with EM algorithm [12] can impute data missing not at random but are restricted to certain parametric models. Deep generative model offers a flexible framework of missing data imputation. For instance, [13, 6, 14] develop variants of recurrent neural networks to impute time series. [8, 9, 7] leverage generative adversarial training (GAN) [15] to learn complex missing patterns. However, all the existing imputation models are autoregressive.

**Non-Autoregressive Modeling**   Non-autoregressive models have gained competitive advantages over autoregressive models in natural language processing [16, 17, 18] and speech [19]. For instance, [19] uses a normalizing flow model [20] to train a parallel feed-forward network for speech synthesis. For neural machine translation, [16] introduce a latent fertility model with a sequence of discrete latent variables. Similarly, [17, 18] propose a fully deterministic model to reduce the amount of supervision. All these works highlight the strength of non-autoregressive models in decoding sequence data in a scalable fashion. Our work is the first non-autoregressive model for sequence imputation tasks with a novel recursive decoding algorithm.

**Generative Adversarial Training**   Generative adversarial networks (GAN) [15] introduce a discriminator to replace maximum likelihood objective, which has sparked a new paradigm of generative modeling. For sequence data, using a discriminator for the entire sequence ignores the sequential dependency and can suffer from mode collapse. [21, 22] develop imitation and reinforcement learning to train GAN in the sequential setting. [21] propose generative adversarial imitation learning to combine GAN and inverse reinforcement learning. [22] develop GAN for discrete sequences using reinforcement learning. We use an imitation learning formula with a differentiable policy.

**Multiresolution Generation**   Our method bears affinity with multiresolution generative models for images such as Progressive GAN [23] and multiscale autoregressive density estimation [24]. The key difference is that [23, 24] only capture spatial multiresolution structures and assume additive models for different resolutions. We deal with multiresolution spatiotemporal structures and generate predictions recursively. Our method is fundamentally different from hierarchical sequence models [25, 26, 27], as it only keeps track of the most relevant hidden states and update them on-the-fly, which is memory efficient and much faster to train.

# 3 Non-Autoregressive Multiresolution Sequence Imputation

Let $X = (x_1, x_2, ..., x_T)$ be a sequence of $T$ observations, where each time step $x_t \in \mathbb{R}^D$. $X$ have missing data, indicated by a masking sequence $M = (m_1, m_2, ..., m_T)$. The masking $m_t$ is zero whenever $x_t$ is missing. Our goal is to replace the missing data with reasonable values for a collection of sequences. A common practice for missing value imputation is to directly model the distribution of the incomplete sequences. One can factorize the probability $p(x_1, \cdots, x_T) = \prod_t p(x_t | x_{<t})$ using chain rule and train a (deep) autoregressive model for imputation [6, 7, 8, 9].

However, a key weakness of autoregressive models is their sequential decoding process. Since the current value is dependent on the previous time steps, autoregressive models often have to resort to sub-optimal beam search and are susceptible to error compounding for long-range sequences [16, 17, 18]. This weakness is worsened in sequence imputation as the model cannot ground the known future, which leads to mismatch between the imputed values and ground truth at the observed points. To alleviate these issues, we instead take a *non-autoregressive* approach and propose a deep, non-autoregressive, multiresolution generative model `NAOMI`.

## 3.1 `NAOMI` Architecture and Imputation Strategy

As shown in Figure 2, `NAOMI` has two components: 1) a forward-backward encoder that maps the incomplete sequences to hidden representations, and 2) a multiresolution decoder that imputes missing values given the hidden representations.

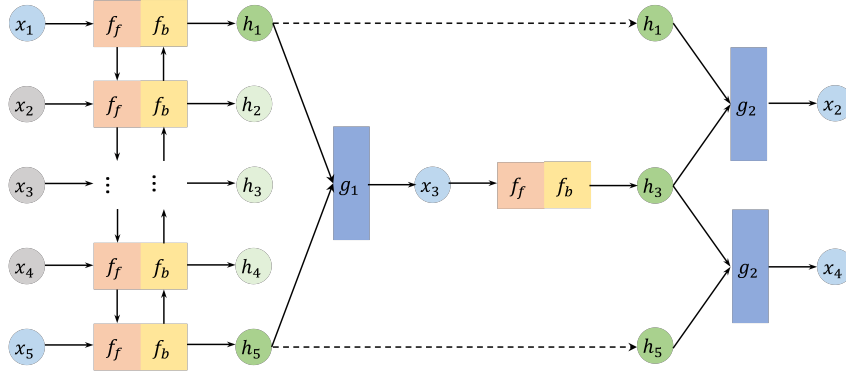

Figure 2: `NAOMI` architecture for imputing a sequence of length five. A forward-backward encoder encodes the incomplete sequence $(x_1, \cdots, x_5)$ into hidden states. The decoder decodes recursively in a non-autoregressive manner: predict $x_3$ using hidden states $h_1, h_5$. After the prediction, the hidden states are updated. Then $x_2$ is imputed based on $x_1$ and the predicted $x_3$, and similarly for $x_4$. This process repeats until all missing values are filled.

**Forward-backward encoder.** We concatenate the observation and masking sequence as input $I = [X, M]$. Our encoder models the conditional distribution of two sets of hidden states given the input: forward hidden states $H^f = (h_1^f, \ldots, h_T^f)$ and backward hidden states $H^b = (h_1^b, \ldots, h_T^b)$:

$$q(H^f | I) = \prod_{t=1}^{T} q(h_t^f | h_{<t}^f, I_{\leq t}) \qquad q(H^b | I) = \prod_{t=1}^{T} q(h_t^b | h_{>t}^b, I_{\geq t}), \qquad (1)$$

where $h_t^f$ and $h_t^b$ are the hidden states of the history and the future respectively. We parameterize the above distributions with a forward RNN $f_f$ and a backward RNN $f_b$:

$$q(h_t^f | h_{<t}^f, I_{\leq t}) = f_f(h_{t-1}^f, I_t) \qquad q(h_t^b | h_{>t}^b, I_{\geq t}) = f_b(h_{t+1}^b, I_t). \qquad (2)$$

**Multiresolution decoder.** Given the joint hidden states $H := [H^f, H^b]$, the decoder learns the distribution of complete sequences $p(X|H)$. We adopt a *divide and conquer* strategy and decode recursively from coarse to fine-grained resolutions. As shown in Figure 2, at each iteration, the decoder first identifies two known time steps as pivots ($x_1$ and $x_5$ in this example), and imputes close to their midpoint ($x_3$). One pivot is then replaced by the newly imputed step and the process repeats at a finer resolution for $x_2$ and $x_4$.

---

**Algorithm 1** Non-Aut**O**regressive **M**ultiresolution **I**mputation

---

1: Initialize generator $G_\theta$ and discriminator $D_\omega$
2: **repeat**
3:     Sample complete sequences from training data $X^* \sim \mathcal{C}$ and mask $M$
4:     Compute incomplete sequences $X = X^* \odot M$
5:     Initialize $h_t^f$, $h_t^b$ using Eqn 2 for $0 \le t \le T$
6:     **while** $X$ contains missing values **do**
7:         Find the smallest $i$ and the smallest $j > i$ s.t. $m_i = m_j = 1$ and $\exists t$, $i < t < j$ s.t. $m_t = 0$
8:         Find the smallest $r$ s.t. $n_r = 2^{R-r} \le (j - i)/2$, thus the imputation point $t^\star = i + n_r$
9:         Decode $x_{t^\star}$ using $p(x_{t^\star}|H) = g^{(r)}(h_i^f, h_j^b)$, update $X$, $M$
10:        Update the hidden states using $h_t^f = f_f(h_{t-1}^f, I_t), \quad h_t^b = f_b(h_{t+1}^b, I_t)$
11:     **end while**
12:     Update generator $G_\theta$ by backpropagation
13:     Train discriminator $D_\omega$ with complete sequences $X^*$ and imputed sequences $\hat{X}$
14: **until** Converge

---

Formally speaking, a decoder with $R$ resolutions consists of a series of decoding functions $g^{(1)}, \ldots, g^{(R)}$, each of which predicts every $n_r = 2^{R-r}$ steps. The decoder first finds two known steps $i$ and $j$ as pivots, and then selects the missing step $t$ that is close to the midpoint: $[(i + j)/2]$. Let $r$ be the smallest resolution that satisfies $n_r \le (j - i)/2$. The decoder updates the hidden states at time $t^\star$ using the forward states $h_i^f$ and the backward states $h_j^b$. A decoding function $g^{(r)}$ then maps the hidden states to the distribution over the outputs: $p(x_t^\star|H) = g^{(r)}(h_i^f, h_j^b)$.

If the dynamics are deterministic, $g^{(r)}$ directly outputs the imputed value. For stochastic dynamics, $g^{(r)}$ outputs the mean and the standard deviation of an isotropic Gaussian distribution, and the predictions are sampled from the Gaussian distribution using the reparameterize trick [28]. The mask $m_t$ is updated to 1 after imputation and the process proceeds to the next resolution. The details of this decoding process are described in Algorithm 1. We encourage the reader to watch our demo video for a detailed visualization and imputed examples. [3]

**Efficient hidden states update.** `NAOMI` efficiently updates the hidden states by reusing the previous computation, which has the same time complexity as autoregressive models. Figure 3 shows an example for a sequence of length nine. Grey blocks are the known time steps. Orange blocks are the target time step to be imputed. Hollow arrows denote forward hidden states updates, and black arrows represent backward hidden states updates. Grey arrows are the outdated hidden states updates. The dashed arrows represent the decoding steps. Earlier hidden states are stored in the imputed time steps and are reused. Therefore, forward hidden states $h^f$ only need to be updated once and backward hidden states $h^b$ are updated at most twice.

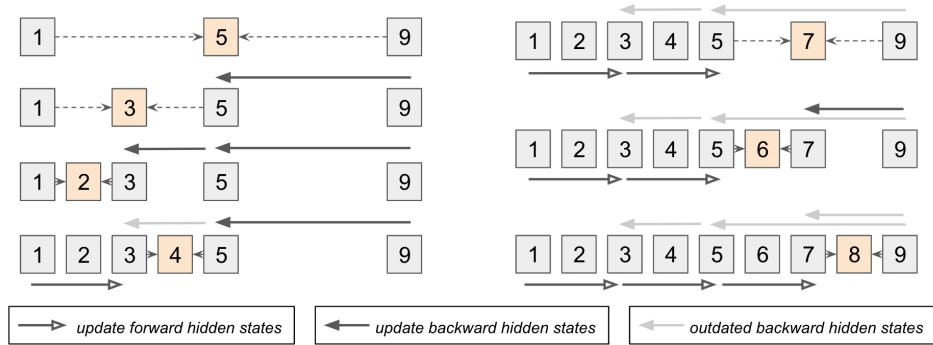

Figure 3: `NAOMI` hidden states updating rule for a sequence of length nine. Note that backward hidden states $h_{9\to7}^b$ are updated twice when predicting $\hat{x}_6$.

**Complexity.** The total run-time of `NAOMI` is $O(T)$. The memory usage is similar to that of bi-directional RNN ($O(T)$), except that we only need to save the latest hidden states for the forward encoder. The decoder hyperparameter $R$ is picked such that $2^R$ is close to the most common missing interval size, and the run time scales logarithmically with the length of the sequence.

## 3.2 Learning Objective

Let $\mathcal{C} = \{X^*\}$ be the collection of complete sequences, $G_\theta(X, M)$ denote our generative model NAOMI parametrized by $\theta$, and $p(M)$ denote the prior over the masking. The imputation model can be trained by optimizing the following objective:

$$\min_\theta \mathbb{E}_{X^* \sim \mathcal{C}, M \sim p(M), \hat{X} \sim G_\theta(X, M)} \left[ \sum_{t=1}^T \mathcal{L}(\hat{x}_t, x_t) \right]. \tag{3}$$

where $\mathcal{L}$ is some loss function. For deterministic dynamics, we use the mean squared error as our loss $\mathcal{L}(\hat{x}_t, x_t) = \|\hat{x}_t - x_t\|_2$. For stochastic dynamics, we can replace $\mathcal{L}$ with a discriminator, which leads to the adversarial training objective. We use a similar formulation as generative adversarial imitation learning (GAIL) [21], which quantifies the distributional difference between generated and training data at the sequence level.

**Adversarial training.** Given the generator $G_\theta$ in NAOMI and a discriminator $D_\omega$ parameterized by $\omega$, the adversarial training objective function is:

$$\min_\theta \max_\omega \mathbb{E}_{X^* \sim \mathcal{C}} \left[ \sum_{t=1}^T \log D_\omega(\hat{x}_t, x_t) \right] + \mathbb{E}_{X^* \sim \mathcal{C}, M \sim p(M), \hat{X} \sim G_\theta} \left[ \sum_{t=1}^T \log(1 - D_\omega(\hat{x}_t, x_t)) \right], \tag{4}$$

GAIL samples the sequences directly from the generator and optimizes the parameters using policy gradient. This approach can suffer from high variance and require a large number of samples [29]. Instead of sampling, we take a model-based approach and make our generator fully differentiable. We apply the reparameterization trick [28] at every time step by mapping the hidden states to mean and variance of a Gaussian distribution.

## 4 Experiments

We evaluate NAOMI in environments with diverse dynamics: real-world traffic time series, billiard ball trajectories from a physics engine, and team movements from professional basketball gameplay. We compare with the following baselines:

- Linear: linear interpolation, missing values are imputed using interpolated predictions from two closest known observations.
- KNN[11]: $k$ nearest neighbours, missing values are imputed as the average of the k nearest neighboring sequences.
- GRUI [9]: autoregressive model with GAN for time series imputation, modified to handle complete training sequence. The discriminator is applied once to the entire time series.
- MaskGAN[7]: autoregressive model with actor-critic GAN, trained using adversarial imitation learning with discriminator applied to every time step, uses a forward encoder only, and decodes at a *single* resolution.
- SingleRes: autoregressive counterpart of our model, trained using adversarial imitation learning, uses a forward-backward encoder, but decodes at a *single* resolution. Without adversarial training, it reduces to BRITS [14].

We randomly choose the number of steps to be masked, and then randomly sample the specific steps to mask in the sequence. Hence the model learns various missing patterns during training. We used the same masking scheme for all methods, including MaskGAN and GRUI. See Appendix for implementation and training details.

### 4.1 Imputing Traffic Time Series

The PEMS-SF traffic time series [30] data contains 267 training and 173 testing sequences of length 144 (sampled every 10 mins throughout the day). It is multivariate with 963 dimensions, representing the freeway occupancy rate collected from 963 different sensors. We generate a masking sequence for each data with 122 to 140 missing values.

**Imputation accuracy** L2 loss between imputed missing values and their ground-truth most accurately measures the quality of the generated sequence. As clearly shown in table 1, NAOMI outperforms others by a large margin, reducing L2 loss by 23% compared to the autoregressive baselines. KNN performs reasonably well, mostly because of the repeated daily traffic patterns in the training data. Simply finding a similar sequence in the training data is sufficient for imputation.

Table 1: Traffic data L2 loss comparison. `NAOMI` outperforms others, reducing L2 loss by **23%** from the autoregressive counterpart.

| Models | NAOMI | SingleRes | MaskGAN | KNN | GRUI | Linear |
|---|---|---|---|---|---|---|
| **L2 Loss** $(10^{-4})$ | **3.54** | 4.51 | 6.02 | 4.58 | 15.24 | 15.59 |

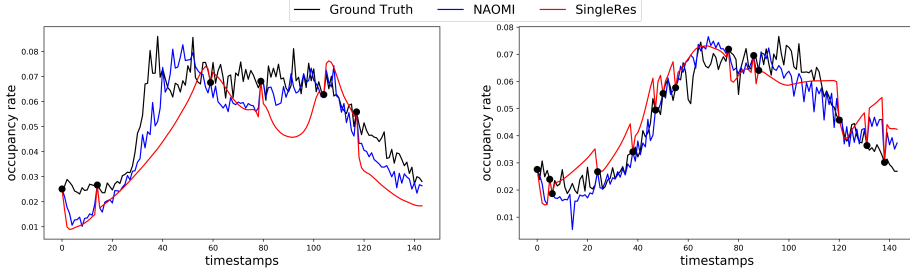

Figure 4: Traffic time series imputation visualization. `NAOMI` successfully captures the multiresolution patterns of the data from observed steps, while `SingleRes` only learns a smoothed version of the original sequence and frequently deviates from ground truth.

**Generated Sequences.** Figure 4 visualizes the predictions from two best performing models: `NAOMI` (blue) and `SingleRes` (red). Black dots are observed time steps and black curves are the ground truth. `NAOMI` successfully captures the pattern of the ground truth time series, while `SingleRes` fails. `NAOMI` learns the multiscale fluctuation rooted in the ground truth, whereas `SingleRes` only learns some averaged behavior. This demonstrates the clear advantage of using multiresolution modeling.

## 4.2   Imputing Billiards Trajectories

We generate 4000 training and 1000 test sequences of Billiards ball trajectories in a rectangular world using the simulator from [31]. Each ball is initialized with a random position and random velocity and rolled-out for 200 timesteps. All balls have a fixed size and uniform density, and there is no friction. We generate a masking sequence for each trajectory with 180 to 195 missing values.

**Imputation accuracy.** Three defining characteristics of the physics in this setting are: (1) moving in straight lines; (2) maintaining unchanging speed; and (3) reflecting upon hitting a wall. Hence, we adopt four metrics to quantify the learned physics: (1) $L_2$ *loss* between imputed values and ground-truth; (2) *Sinuosity* to measure the straightness of the generated trajectories; (3) *Average step size change* to measure the speed change of the ball; and (4) *Distance between reflection point and the wall* to check whether the model has learned the physics underlying collision and reflection.

Comparison of all models w.r.t. these metrics are shown in Table 2. Expert represents the ground truth trajectories from the simulator. Statistics closer to the expert are better. We observe that `NAOMI` has the best overall performance across almost all the metrics, followed by `SingleRes` baseline. It is expected that `linear` to perform the best w.r.t step change. By design, linear interpolation maintains a constant step size change that is the closest to the ground-truth.

**Generated trajectories.** We visualize the imputed trajectories in Figure 5. There are 8 known timesteps (black dots), including the starting position. `NAOMI` can successfully recover the original trajectory whereas `SingleRes` deviates significantly. Notably, `SingleRes` mistakenly predicts the

Table 2: Metrics for billiards imputation accuracy. Statistics closer to the expert indicate better model performance. `NAOMI` has the best overall performance, reducing deviation from ground truth by **30% to 70%** across all metrics compared to autoregressive baselines.

| Models | Linear | KNN | GRUI | MaskGAN | SingleRes | NAOMI | Expert |
|---|---|---|---|---|---|---|---|
| **Sinuosity** | 1.121 | 1.469 | 1.859 | 1.095 | 1.019 | **1.006** | 1.000 |
| **step change** $(10^{-3})$ | **0.961** | 24.59 | 28.19 | 15.35 | 9.290 | 7.239 | 1.588 |
| **reflection to wall** | 0.247 | 0.189 | 0.225 | 0.100 | 0.038 | **0.023** | 0.018 |
| **L2 loss** $(10^{-2})$ | 19.00 | 5.381 | 20.57 | 1.830 | 0.233 | **0.067** | 0.000 |

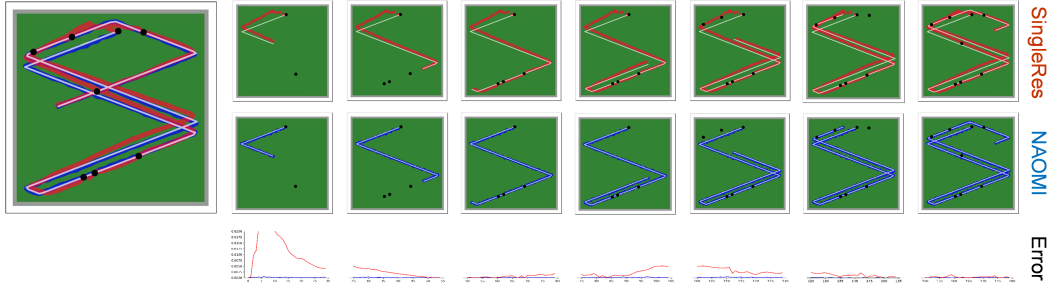

Figure 5: Comparison of imputed billiards trajectories. Blue and red trajectories/curves represent `NAOMI` and the single-resolution baseline model respectively. White trajectories represent the ground-truth. There are 8 known observations (black dots). `NAOMI` almost perfectly recovers the ground-truth and achieves lower stepwise L2 loss of missing values than the baseline model (third row). The trajectory from the baseline first incorrectly bounces off the upper wall, which results in curved paths that deviate from the ground-truth.

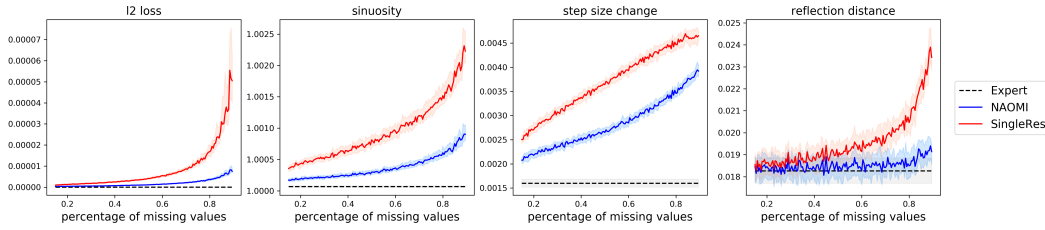

Figure 6: Billiards model performance with increasing percentage of missing values. The median and 25, 75 percentile values are displayed at each number of missing steps. Statistics closer to the expert indicate better performance. `NAOMI` performs better than `SingleRes` for all metrics.

ball to bounce off the upper wall instead of the left wall. As such, `SingleRes` has to correct its behavior to match future observations, leading to curved and unrealistic trajectories. Another deviation can be seen near the bottom-left corner, where `NAOMI` produces trajectory paths that are truly parallel after bouncing off the wall twice, but `SingleRes` does not.

**Robustness to missing proportion.** Figure 6 compares the performance of `NAOMI` and `SingleRes` as we increase the proportion of missing values. The median value and 25, 75 percentile values are displayed for each metric. As the dynamics are deterministic, higher missing portion usually means bigger gaps, making it harder to find the *correct* solutions. We can see both models' performance degrade drastically as we increase the percentage of missing values, but `NAOMI` still outperforms `SingleRes` in all metrics.

### 4.3 Imputing Basketball Players Movement

The basketball tracking dataset contains the trajectories of professional basketball players on offense with 107,146 training and 13,845 test sequences. Each sequence contains the (x, y)-coordinates of 5 players for 50 timesteps at 6.25Hz and takes place in the left half-court. We generate a masking sequence for each trajectory with 40 to 49 missing values.

Table 3: Metrics for basketball imputation accuracy. Statistics **closer to the expert** indicate better model performance. `NAOMI` has the best overall performance, reducing deviation from ground truth by **more than 70%** compared to autoregressive baselines.

| Models | Linear | KNN | GRUI | MaskGAN | SingleRes | NAOMI | Expert |
|---|---|---|---|---|---|---|---|
| **Path Length** | 0.482 | 0.921 | 1.141 | 0.793 | 0.702 | **0.573** | 0.556 |
| **OOB Rate** $(10^{-3})$ | 2.997 | **0.128** | 4.703 | 4.592 | 3.874 | 1.733 | 0.861 |
| **Step Change** $(10^{-3})$ | 0.522 | 13.24 | 14.95 | 9.622 | 4.811 | **2.565** | 1.982 |
| **Path Difference** | 0.519 | 0.746 | 0.690 | 0.680 | 0.571 | **0.581** | 0.580 |
| **Player Distance** | 0.422 | 0.403 | 0.398 | 0.427 | 0.417 | **0.423** | 0.425 |

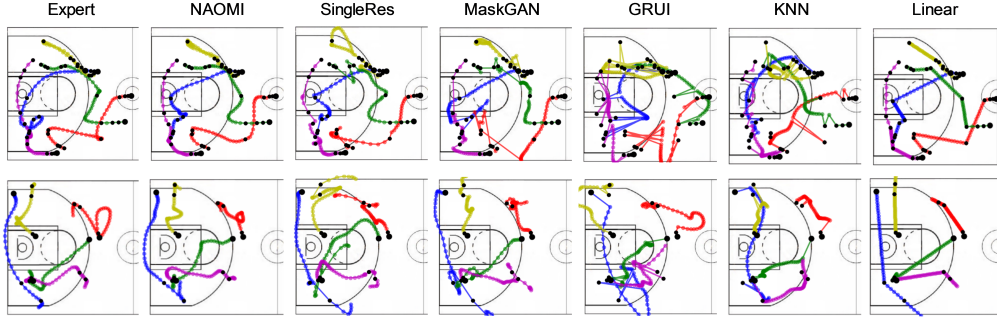

Figure 7: Comparison of imputed basketball trajectories. Black dots represent known observations (10 in first row, 5 in second). Overall, NAOMI produces trajectories that are the most consistent and have the most realistic player velocities and speeds.

**Imputation accuracy.** Since the environment is stochastic (basketball players on offense aim to be unpredictable), measuring L2 loss between our model output and the ground-truth is not necessarily a good indicator of realistic trajectories [32, 33]. Hence, we follow previous work and compute domain-specific metrics to compare trajectory quality: (1) *Average trajectory length* to measure the typical player movement in 8 seconds; (2) *Average out-of-bound rate* to measure the odds of trajectories going out of court boundaries; (3) *Average step size change* to quantify the player movement variance; (4) *Max-Min path diff*; and (5) *Average player distance* to characterize the team coordination. Table 3 compares model performances using these metrics. Expert represents real human play, and the closer to the expert data, the better. NAOMI outperforms baselines in almost all the metrics.

**Generated trajectories.** We visualize imputed trajectories from all models in Figure 7. NAOMI produces trajectories that are the most consistent with known observations and have the most realistic player velocities and speeds. In contrast, other baseline models often fail in these regards. KNN generates trajectories with unnatural jumps as finding nearest neighbors becomes difficult with dense known observations. Linear fails to generate curvy trajectories when few observations are known. GRUI generates trajectories that are inconsistent with known observations. This is largely due to mode collapse caused by applying a discriminator to the entire sequence. MaskGAN, which relies on seq2seq and a single encoder, fails to condition on the future observations and predicts straight lines.

**Robustness to missing proportion.** Figure 8 compares the performance of NAOMI and SingleRes as we increase the proportion of missing values. The median value and 25, 75 percentile values are displayed for each metric. Note that we always observe the first step. Generally speaking, more missing values make the imputation harder, and also brings more uncertainty to model predictions. We can see that performance (average performance and imputation variance) of both models degrade with more missing values. However, at a certain percentage of missing values, the performance of imputation starts to improve for both models.

This shows an interesting trade-off between *available information* and *number of constraints* for generative models in imputation. More observations provide more information regarding the data distribution, but can also constrain the learned model output. As we reduce the number of observations, the model can learn more flexible generative distributions, without conforming to the constraints imposed by the observed time steps.

**Learned conditional distribution.** Our model is fully generative and learns the conditional distribution of the complete sequences given observations. As shown in Figure 9. For a given set of known observations, we use NAOMI to impute missing values with 50 different random seeds and overlay the generated trajectories. We can see that as the number of known observations increases, the variance of the learned conditional distribution decreases. However, we also observe some mode collapse in our model: the trajectory of the purple player in the ground truth is not captured in the conditional distribution in the first image.

## 4.4 Forward Prediction

Forward prediction is a special case of imputation when all observations, except for a leading sequence, are missing. We show that NAOMI can also be trained to perform forward prediction without

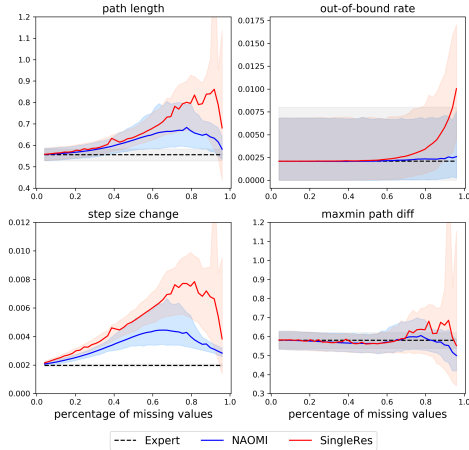

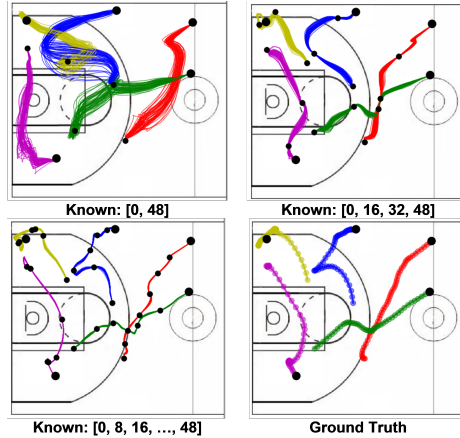

Figure 8: Basketball model performance with increasing percentage of missing values. The median and 25, 75 percentile values are displayed. Statistics closer to the expert indicate better performance. `NAOMI` performs better than `SingleRes` for all metrics.

Figure 9: The generated conditional distribution of basketball trajectories given known observations (black dots) with sampled trajectories. As the number of known observations increases, the variance of the predictions, hence the model uncertainty decreases.

modifying the model structure. We take a trained imputation model as initialization, and continue training for forward prediction by using the masking sequence $m_i = 0, \forall i \geq 5$ (first 5 steps are known). We evaluate forward prediction performance using the same metrics.

Figure 10 compares forward prediction performance in Billiards. Without any known observations in the future, autoregressive models like `SingleRes` are effective in learning consistent step changes, but `NAOMI` generates straighter lines and learns the reflection dynamics better than other baselines.

| Models | RNN | SingleRes | NAOMI | Expert |
|---|---|---|---|---|
| **Sinuosity** | 1.054 | 1.038 | **1.020** | 1.00 |
| **Step Change** $(10^{-3})$ | 11.6 | **9.69** | 10.8 | 1.59 |
| **Reflection to wall** | 0.074 | 0.068 | **0.036** | 0.018 |
| **L2 Loss** $(10^{-3})$ | 4.698 | 4.753 | **1.682** | 0.0 |

Figure 10: Billiard Forward Prediction Comparison. Top: metrics for billiards prediction accuracy. Statistics **closer to the expert** indicate better model performance. Bottom: predicted billiards trajectories. Black dots represent known observations. `NAOMI` perfectly recovers the ground-truth.

## 5 Conclusion

We propose a deep generative model `NAOMI` for imputing missing data in long-range spatiotemporal sequences. `NAOMI` recursively finds and predicts missing values from coarse to fine-grained resolutions using a non-autoregressive approach. Leveraging multiresolution modeling and adversarial training, `NAOMI` is able to learn the conditional distribution given very few known observations. Future work will investigate how to infer the underlying distribution when complete training sequences are not available. The trade-off between partial observations and external constraints is another direction for deep generative imputation models.

**Acknowledgments.** This work was supported in part by NSF #1564330, NSF #1850349, and DARPA PAI: HR00111890035.

## Footnotes

[2]`https://github.com/felixykliu/NAOMI`

[3] https://youtu.be/eoiK42w02w0

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
