[Supplementary Material]

# 1   Appendix

## A.   Model details

The forward and backward encoders are both 2-layer RNNs with GRU cells. The multiresolution decoder consists of multiple 2-layer fully-connected neural networks. For the adversarial training, we use a 1-layer RNN with GRU cells as the discriminator. We train on squared loss for billiards and traffic data, and adversarial loss for basketball. Our submitted code contains more details about other hyper-parameters, like learning rate, learning rate decay, and adversarial training strategy. All evaluation results (except for separately described) are computed from 500 runs with batch size 64.

Table 1 lists the hyper-parameters of our model. Table 2 lists the hyper-parameters of the baselines.

| | $R$ | RNN size | # of params |
|---|---|---|---|
| Basketball | 4 | 275 | 1,842,055 |
| Billiards | 5 | 200 | 1,130,810 |
| Traffic | 4 | 300 | 2,629,700 |

Table 1: `NAOMI` hyperparameters. Our multiresolution decoder has $R$ levels. RNN size applies for both encoder and decoder.

| | | RNN size | # of params |
|---|---|---|---|
| Basketball | SingleRes | 300 | 1,832,420 |
| | MaskGAN | 300 | 1,742,420 |
| Billiards | SingleRes | 230 | 1,067,662 |
| | MaskGAN | 230 | 1,014,762 |
| Traffic | SingleRes | 340 | 2,606,380 |
| | MaskGAN | 340 | 2,014,762 |

Table 2: Hyperparameters of baseline models.

For deterministic dynamics (traffic and Billiards), we use the $L_2$ loss (teacher forcing is applied during pretraining). For stochastic dynamics (e.g. Basketball), we use GAN loss first pretrain the generator using cross-entropy loss for supervised, and then optimize the generator and discriminator alternatively using the training objective in Eqn 5.

## B.   Billiards stats with error bars

Figure 1: Metrics for billiards imputation accuracy. The average value and 5, 95 percentile values are displayed for each metric. Y-axis is splitted to focus on the comparison between `NAOMI` and `SingleRes`. The black thick horizontal lines are the ground truth stats. Statistics closer to the black lines indicate better model performance. `NAOMI` has the best overall performance, reducing deviation from ground truth by **30% to 70%** across all metrics compared to autoregressive baselines.

## C. Model performance with change of model capacity

Figure 2 shows the comparison of billiard trajectory L2 loss between `NAOMI` and `SingleRes` with respect to the total number of parameters from 500 random runs. We can see that `NAOMI` is much more parameter-efficient than the single resolution baseline. Surprisingly, the smallest multiresolution model is more accurate than the largest single resolution baseline.

Figure 2: Billiards L2 loss of different models with different sizes. Error bar here is the *std* of L2 loss, which represents the stability of the model. Our multiresolution model is much more stable and parameter-efficient than the baseline model.

## D. Forward inference visualization

Figure 3: Billiard forward inference comparison

Figure 3 shows the generated trajectories from forward prediction. We can clearly see that `NAOMI` generate much better trajectories.

## E. Theoretical Justification

The design of `NAOMI` draws inspiration from wavelet theory **?**. A sequence $f(x_1, x_2, \cdots, x_T)$ can be approximated by its multiresolution components at $R$ levels, that is $f(x) \approx f_R(x) = \sum_{r=1}^{R} g^{(r)}(x)$. $g^{(1)}, g^{(2)}, \cdots, g^{(R)}$ from a set of nested vector spaces $V_1 \subset V_2 \cdots, \subset V_r, \cdots \subset V_R$ that satisfy: These functions satisfy the following conditions and the approximation error becomes progressively smaller as resolution increases.

The following proposition states the approximation power of the multiresolution decoder:

**Proposition 1.1** *The approximation error of the multiresolution decoder decreases exponentially with the number of resolutions:*

$$g^{(r)} = f(x_1, x_{n_r}, x_{2n_r}, \cdots), \quad g^{(r)} \in V_r$$

*with each decoder approximates the function $g^{(r)}(x)$*