[Reviews · NeurIPS 2019]

Reviewer 1



This paper introduces a novel approach called NAOMI for imputation of arbitrary missing values in dynamic time series. NAOMI fills in missing values in a multi-resolution fashion from recursively coarse to grain, such as first the beginning and then the end and then the middle of the two, and then the middle of the left-half, sharing weights for gaps of the same size. The architecture is "iterative" from coarse to grain but the generation is non-autoregressive (using the re-parameterization trick for cases that are stochastic), allowing for the generator to be fully differentiable and adversarial training to be applied. The paper presents improved empirical results on multiple datasets, ranging from real-world traffic time series, billiard ball trajectories from a physics engine and team movements from professional basketball gameplay. Multiple competitive baselines are being compared to, allowing one to compare results from modifying different components of the model design such as autoregressive versus non-autoregressive, with or without adversarial training. Strengths - The method is novel, the recursive divide and conquer formulation is well-suited for capturing long-term dependencies. Figure 3 shows a clear case for the advantages, both qualitatively and quantitatively. Also, the procedure is quite efficient, memory for hidden states are linear to sequence length while the recursive (shared among similar gap lengths) imputation weights scales logarithmically with sequence length. - The datasets are challenging and cover a wide range of tasks. In the case of traffic time series (4.1), the data is very high dimensional (963 different sensors). Multiple metrics, in additional to L2 loss, are used to capture different aspects of the time-series, accounting for invariant surface variations. - The visualizations are very helpful in understanding the complexity of the tasks and also for comparing the strengths and weaknesses of different methods. - The paper is well-written. - The proposed method is clearly superior in all tasks. Potential weaknesses - Motivation: how does modeling these time series vary or similar to modeling audio, images, or text? The paper mentions time series are more dynamic, involving temporal irregularities such as discontinuities and varying levels of rate of change in different regions, perhaps motivate a bit more why a fixed “sampling” scheme of "halving" the sequence addresses the challenges in the domain or other considerations for future work. - The method assumes fixed-length sequences. Is this common in time series problems and datasets? - How much are the given anchor points (distributed across the sequence) giving the long-term structure versus the model learning them? - [masking] The masking scheme is not described. The figures seem to show that the unmasked positions are those in order with how the divide-and-conquer scheme would proceed. Does make it harder for certain baselines to cope with? Is the masking per time step, that is all dimensions within a time step is masked if a tilmestep is masked? One of the baselines, the GRUI paper, uses random masking across both time steps and dimensions. Given the divide-and-conquer scheme it might not be directly applicable, but would a variation of it be? One question the reader might have is that if for example if the unmasked positions are less well distributed throughout the sequence, how would it affect the performance? In line 235 “Robustness to percentage of missing values”, the word “percentage” could be a bit misleading because it’s not a random percentage but portion in a pre-determined ordering. - [baseline] Would another baseline be generating auto-regressively but using the divide and conquer ordering, without adversarial loss? - Out of scope for this paper, perhaps question for future work, would some modification of the Transformer architectures be a competitive candidate for capturing these long-term dependencies? Clarifying questions: - In the complexity section (line 139), why is only the backward hidden states updated when the step is imputed and not he forward states also. - From the description of the text (line 131 to 133), would line 10 (would include how g^{(r)} is obtained?) and 11 be swapped? - In Figure 2, would g_1 and g_2 be reversed since the subscript r refers to the resolution (size of gap)?

Reviewer 2



I would like to provide weak points of this paper. 1. The statement "Given the imputation algorithm, the forward RNN only encodes complete sequences, while the backward RNN needs to deal with missing values." is too short of making the proposed forward-backward encoder from a bi-directed RNN. Fig. 2 seems to feed all data to f_f and f_b encoders. 2. Based on the result of MaskGan and GRUI, the PEMS-SF traffic time series might be too small to train deep models requiring more than ten thousands samples. Other non-linear but straightforward interpolation methods, such as bicubic spline, would make the author's argument more concrete. Table 1 and 2 report the L2 loss for checking the interpolation accuracy, but somehow, Table 3 does not. Moreover, does Linear showed smaller errors than NAOMI on Path Lengths, Step Change, Path Different, and Player Distance?

Reviewer 3



1) The proposed model might not address the error propagation issue for long-range sequence imputation tasks as effectively as claimed. The imputation process highly relies on the predictions made at the coarsest level (r=1), since the predicted value will be encoded and utilized to update hidden states. The coarse-level imputation tasks, meanwhile, are relatively difficult because of two factors: first, the available information at this stage is the incomplete sequence only, unlike fine-level imputation tasks where we could utilize former predictions; second, the imputation task is formulated to estimate the value of a 'midpoint' and the skipped time-steps at this stage is large from both sides. Thus, the predictions at this level should be considered noisy, and bring negative influences on further imputations that treat equally on the observed and predicted values. The training of the decoder at this level also raises some concerns, which is specified in the next bullet point. 2) The training of multiple decoders might be insufficient. As discussed above, coarse-level decoders (i.e., g^r with smaller r) are of great importance. However, the training of decoders at the coarse levels are not sufficient due to 1) the number of training instances are sparse for a large size gap; and 2) decoders are optimized individually. Also, although the decoded results are further utilized (e.g., g_1 -> x_3 -> h_3 in Fig.1), it is not specified in the paper whether gradients are back-propagated from these paths and it is non-trivial to fulfill this. One interesting potential would be realizing some sharing of parameters between decoders at different levels. 3) There are also some concerns about experiment settings and results analysis. The selection between maximum likelihood and adversarial training object is not clear. It is stated that two objects are for deterministic (Traffic and Billiards datasets) and stochastic (Basketball dataset) settings, respectively. However, it seems straightforward to use L2-loss based training object for the Basketball dataset and vice versa. It would be better to have discussions of training objects selection or experimental comparison. The evaluation metrics for experiments on Basketball dataset are all indirect measurements. Considering the ground-truth position trajectories are provided in the dataset as (x, y)-coordinates, it's better to include the L2-loss of different approaches. Besides Figure 6, it will make the experiment results more convincing if various percentages of missing values are studied for more datasets. For experiments on Traffic and Billiards datasets, the presented results are only when a large amount of data is missing (e.g., 122 out of 144, 180 out of 200) Some minor comments: - In Algorithm 1, the step of updating mask m_t after imputation is missing, and recursive imputation is not presented clearly. - In Eqn 3, the choice of 'pivots' (at t-n_r, t+n_r) is not aligned with the description, i.e., using observed or predicted values. - What are the input representations of missing values for encoders? - The plots of error in Fig.4 are not very well readable.

[Author Response · NeurIPS 2019]

We thank the reviewers for their insightful comments. We first clarify our approach and then address specific concerns.

**R1, R2** *Forward-backward asymmetry and decoding strategy.* NAOMI efficiently uses forward and backward hidden
states ($h^f, h^b$). Note that encoder and decoder share weights. For example, consider a situation where only $x_0$ and
$x_8$ are known, and we wish to impute $x_{1\rightarrow 7}$ ($x_1$ to $x_7$). We first predict the mid-point $\hat{x}_4 = g(h_0^f, h_8^b)$ and update the
backward hidden states $h_{8\rightarrow 4}^b$. Given $\hat{x}_4$, we predict $\hat{x}_2 = g(h_0^f, h_4^b)$ and update $h_{4\rightarrow 2}^b$. Recursively, given $x_0, \hat{x}_2$, we
predict $\hat{x}_1$. Since $x_{0\rightarrow 2}$ are now known or imputed, we update the forward states $h_{0\rightarrow 2}^f$ and predict $\hat{x}_3 = g(h_2^f, h_4^b)$. For
the second half, after predicting $\hat{x}_6 = g(h_4^f, h_8^b)$, we update $h_{4\rightarrow 6}^f$ and $h_{8\rightarrow 6}^b$. *Note that $h_{8\rightarrow 6}^b$ have been updated before*
*when predicting $\hat{x}_4$!* More generally, the forward states $h^f$ are updated once whereas the backward states $h^b$ are twice.
We encourage the reviewers to check the supplementary material, with code and visualizations of our decoding strategy.

**R2, R3** *Evaluation metrics.* Evaluating generative models is an open problem, e.g., log-likelihood does not correlate
well with generation quality [Theis et al., 2015]. In our case, neither L2 nor log-likelihood can capture "realistic"
player behavior in basketball [Zheng et al., 2016, Zhan et al., 2019]. Hence, we follow previous work and compute
domain-specific metrics (speed, distance traveled, out-of-bounds rate) to compare trajectory quality. We will include
L2-loss for the basketball dataset, but note that NAOMI (*0.013*) still outperforms SingleRes (*0.040*).

**R1** *Motivation.* In general, time-series data features different types of dynamics and missing value patterns compared
to text and images. Time-series data are often multi-resolution, which are exploited by our model via the divide-and-
conquer strategy. Note we do not use a fixed sampling scheme for missing values (see below). We would consider
combining NAOMI with convolutional or Transformer-based approaches to handle high-dimensional sequences.

*Fixed length.* No, our method does *not* assume fixed-length sequences. NAOMI can decode and train on varying-length
sequences, e.g., by padding shorter sequences to a maximal length.

*Masking.* We mask all dimensions for $n$ randomly chosen time-steps, which is independent of the order of the divide-
and-conquer strategy (see Algo. 1). We used the same masking scheme for all methods, including `MaskGAN` and `GRUI`.
Note the "halving" scheme in Figure 2 is only an example: NAOMI is compatible with *any* masking pattern. If the
second half of a sequence is masked, NAOMI is pure forward inference (see supplemental material for results).

*Auto-regressive baseline with divide-and-conquer.* Note that "auto-regressive" and "divide-and-conquer" are mutually
exclusive decoding strategies, hence this baseline does not exist by definition.

*Transformer.* We agree that applying NAOMI to Transformer models is interesting, but leave this for future work.

**R2** *a bi-directed RNN . . . to $f_f$ and $f_b$ encoders.* NAOMI iteratively (re)-encodes and decodes as described above. Only
the initial sequence encoding (see Figure 2) behaves like a bi-directional RNN. *Bi-cubic spline.* We are happy to add
this, but since we compared with the state-of-the-art baselines (e.g., GRUI) for sequence imputation, we believe our
results stand on their own. *Table 3, does* `Linear` *show smaller errors.* The results in Table 3 are *not* errors, but *sample*
*statistics*, i.e., the closer to the Expert, the better. `Linear` has smaller values, but are actually worse as they are further
away from the Expert statistics. We chose these metrics for the reasons explained above.

**R3** *. . . might not address error propagation . . .* We agree that `NAOMI` may not fully solve error propagation for "any"
gap size between observed time steps. However, we compared many model variations and baselines on three time-series
datasets with different sequence lengths and varying missing-value proportions. Our extensive experiments have
empirically shown the effectiveness of NAOMI. We believe noisy coarse predictions are not an issue on these datasets
mostly due to the multiresolution structure and spatiotemporal smoothness of the data. Hence, even noisy coarse-level
predictions provide reasonable grounding points at finer resolutions.

*Multiple decoders might be insufficient* There appear to be several misunderstandings. Note that we randomly sample
masking patterns for each training step. Hence the model will see gaps of varying sizes during training. We compute
the loss at the sequence level to make sure the entire sequence look real, which co-trains the decoders. We repeatedly
use the reparameterization trick (L136) to make every sampling operation and hence the entire imputation procedure
differentiable. Our experimental results demonstrate this end-to-end training approach is sufficient for our datasets.

*Maximum likelihood and adversarial training* NAOMI is a non-autoregressive generator, *which can be trained with any*
*objective*. We used L2-loss for traffic and billiards because it is standard in those domains, see explanation above.

*Missing values . . . datasets.* We will add analyses similar to Figure 6 for all datasets to the Appendix.

*Minor comments.* We will make Figure 4 more readable. For inputs of the backward encoder, the first dimension is 1
for known or predicted steps, 0 for masked steps, and the other dimensions are 0 for masked steps.

[1] E. Zhan, et al. (2019), "Generating Multi-Agent Trajectories using Programmatic Weak Supervision," *ICLR*, 2019

[2] S. Zheng, et al. (2016), "Generating long-term trajectories using deep hierarchical networks," *NIPS*, 2016


[Meta-Review · NeurIPS 2019]

This paper proposes a compelling new model for doing sequence imputation non-autoregressively. The divide and conquer strategy seems to be very effective and there is a thorough quantitative and qualitative evaluation.